# Probabilistic Machine Learning Methods for Fractional Brownian Motion Time Series Forecasting

Lyudmyla Kirichenko [1,*] and Roman Lavrynenko [2]

1 Applied Mathematics Department, Kharkiv National University of Radio Electronics, 61166 Kharkiv, Ukraine
2 Artificial Intelligence Department, Kharkiv National University of Radio Electronics, 61166 Kharkiv, Ukraine; roman.lavrynenko.cpe@nure.ua
* Correspondence: lyudmyla.kirichenko@nure.ua; Tel.: +380-506601370

**Abstract:** This paper explores the capabilities of machine learning for the probabilistic forecasting of fractional Brownian motion (fBm). The focus is on predicting the probability of the value of an fBm time series exceeding a certain threshold after a specific number of time steps, given only the knowledge of its Hurst exponent. The study aims to determine if the self-similarity property is preserved in a forecasting time series and which machine learning algorithms are the most effective. Two types of forecasting methods are investigated: methods with a predefined distribution shape and those without. The results show that the self-similar properties of the fBm time series can be reliably reproduced in the continuations of the time series predicted by machine learning methods. The study also provides an experimental comparison of various probabilistic forecasting methods and their potential applications in the analysis and modeling of fractal time series.

**Keywords:** fractional Brownian motion; hurst exponent; time series; probabilistic forecast; quantile regression; distributional modelling

## 1. Introduction

Fractional Brownian motion (fBm) is a stochastic process characterized by long-range dependence, self-similarity, and non-stationarity. These properties make fBm a suitable framework for modeling time series data with complex structures and memory effects, such as financial market data, Internet traffic, realizations of biomedical signals, dynamic indicators of natural phenomena, etc. [1–5]. Predicting the future behavior of stochastic fractal time series can help make more informed decisions. For instance, in finance, forecasts enable risk assessment and decision making regarding an asset purchase or sale. Predicting Internet traffic allows for optimal scaling of network resources, and anticipation of peak loads, ensuring sufficient bandwidth for high-quality user service. Moreover, forecasting fractal time series of seismic activity can provide insights into potential trends and probabilities of earthquakes, among other applications. These predictions contribute to a wide range of fields and can lead to more effective decision-making processes [6–10]. Research into fractional processes has the potential to advance emerging directions in computer science, such as high-order fractional complex-valued bidirectional associative memory neural networks with multiple time delays [11–13].

Despite the widespread usage of point forecasting and interval forecasting methods in time series analysis, these approaches alone prove insufficient for stochastic time series such as fBm. The application of probabilistic forecasting confers a range of advantages. Probabilistic forecasting provides a more comprehensive representation of future outcomes by estimating the entire probability distribution of the forecast variable, rather than just a single value.

Probabilistic forecasting methods have become increasingly important in various fields, such as finance, weather, and energy. These methods provide valuable information on the

distribution of future outcomes, which is crucial for decision making and risk assessment. This information can be used to assess the risks associated with different decisions and to implement risk management strategies.

There is a multitude of research related to the forecasting of self-similar time series but the majority of existing methods primarily focus on point and interval forecasting methods [14–16]. However, the challenge of probabilistic forecasting for fractal time series, particularly fBm, remains unresolved. This highlights the need for further research and development of techniques that address probabilistic forecasting in the context of fractal time series.

Probabilistic forecasting methods can be broadly categorized into two classes based on the underlying distribution shape assumptions: (1) methods with a predefined distribution shape and (2) methods without a predefined distribution shape. The key difference between these two categories is the degree of flexibility in modeling the underlying uncertainty:

1.  Methods that assume parameters of a specific probability distribution that the forecast variable follows, such as the Gaussian, Gamma, or Weibull distribution. These methods are characterized by a finite set of parameters that are estimated from the data. Examples of parametric methods include Natural Gradient Boosting for Probabilistic Prediction (NGBoost) [17], XGBoostLSS and LightGBMLSS, which model all moments of a parametric distribution: mean, location, scale, and shape (LSS) [18,19], Catboost with Uncertainty (CBU), Probabilistic Gradient Boosting Machines (PGBM) [20], and Instance-Based Uncertainty Estimation for Gradient-Boosted Regression Trees (IBUG) [21].
2.  Methods that estimate the distribution directly from the data without relying on any predefined shape, often focusing on predicting quantiles. These methods provide increased flexibility when modeling complex distributions, making them especially valuable when the true distribution is unknown or significantly deviates from standard parametric forms. Quantile regression is a popular nonparametric technique [22,23]. The newest approach is Nonparametric Probabilistic Regression with Coarse Learners (PRESTO) [24].

The objective of this research comprises two stages. The first stage focuses on exploring the potential for long-term probabilistic forecasting of fBm realizations while maintaining the Hurst exponent. In this stage, fBm realizations with various Hurst exponent values will be modeled using the Hosking algorithm [25], and forecasted realizations (continuations) will be obtained using several primary probabilistic forecasting methods. These continuations will be examined for preserving the Hurst exponent value using the Whittle estimator and normal distribution using the Anderson–Darling test [26]. The preservation of the Hurst exponent, normal distribution, and root mean square increments of fBm realizations and their continuations allows us to conclude that the properties of fBm realizations are preserved when forecasting with machine learning methods.

The second stage is directed toward conducting an experimental comparison of various methods for predicting fBm realizations. In this part of the research, both parametric and nonparametric probabilistic forecasting methods employing machine learning will be considered. Computational experiments will be carried out, where each method will be applied to forecast fBm realizations with different Hurst exponent values. The accuracy and efficiency of each method will then be compared to determine the most suitable forecasting method for these types of data.

Our Contribution:

1.  We demonstrate that the self-similar properties of the fBm time series can be reliably reproduced in the continuations of the series predicted by machine learning methods.
2.  We propose an approach for evaluating probabilistic methods in forecasting the fBm time series. Our method involves efficiently computing multiple continuations of a single fBm time series using the Hosking algorithm, resulting in a dataset with ground truth quantiles of possible continuations. Using a one-period-ahead model, iterated forward for the desired number of periods, we can predict quantiles of continuations

of fBm time series over hundreds or thousands of time steps and compare them with the ground truth quantiles (Figure 1).

3. We conducted an experimental comparison of various probabilistic forecasting methods using this approach, providing insights into their performance and potential applications in the analysis and modeling of fractal time series.

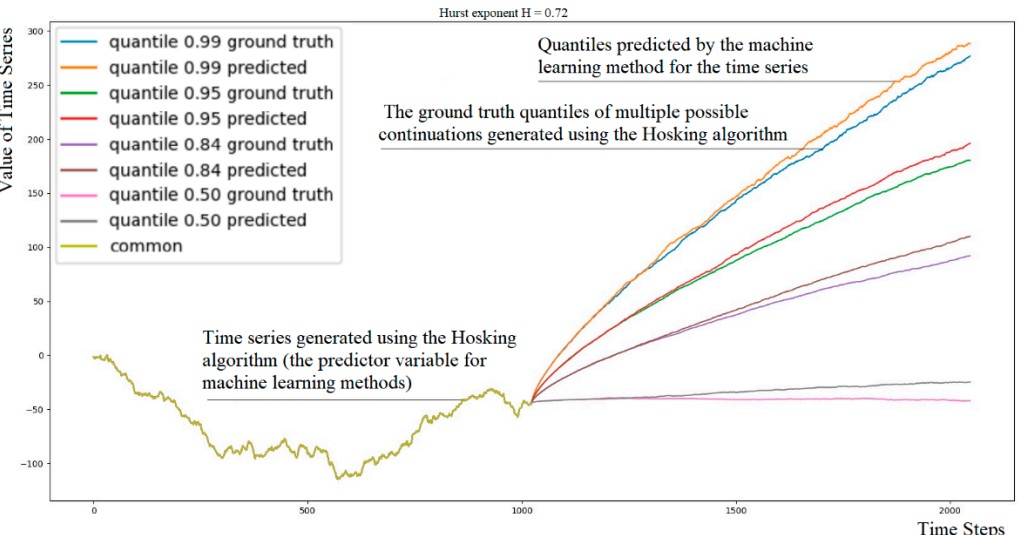

**Figure 1.** Summary of our approach. We compute multiple continuations of a single fBm time series using the Hosking algorithm, resulting in a dataset with ground truth quantiles of possible continuations. Using a one-period-ahead model, iterated forward for the desired number of periods, we can predict quantiles of continuations for fBm time series over hundreds or thousands of time steps. By comparing the predicted quantiles with the ground truth quantiles, we can evaluate the performance of the chosen machine learning method.

## 2. Materials and Methods

### 2.1. Problem Setting

To evaluate the performance of a probabilistic forecasting model, it is necessary to compare the predicted probability distribution with the actual probability distribution of the ground truth data. One common way to do this is by calculating the quantiles of the predicted probability distribution and comparing them with the quantiles of the actual distribution.

Predicting only one-time step ahead is insufficient for modeling the behavior of a time series with fractal properties in the future and evaluating the method's performance. It is necessary to construct sufficiently long continuations of the time series to assess the presence of similar fractal properties in these continuations. To achieve this, we used iterated approaches that utilize one-step-ahead prediction models, with multi-step predictions obtained by recursively feeding predictions into future inputs. This allows us to generate longer continuations of the time series while preserving the fractal characteristics, which can then be compared to the ground truth continuations for method evaluation and performance analysis.

Additionally, predicting merely one continuation of a fractal time series is not enough to evaluate the method. It is crucial to generate a representative number of possible time series continuations and compare the quantiles of the obtained continuations for each time step with the quantiles of the true possible continuations.

Our research will focus on the probabilistic forecasting of trajectories for a self-similar process. A process $\{X(t), t \in R\}$ is said to be self-similar of Hurst exponent H, if, for any $a > 0$:

$$\text{Law}\{X(at)\} = \text{Law}\{a^H\{X(t)\}, \tag{1}$$

that is the equality of n-dimensional distribution laws of X(t) [27,28].

There are several approaches to defining fractional Brownian motion [29,30], and here we apply one of them [28] that emphasizes the properties of normality and self-similarity. Fractional Brownian motion X(t) is a type of self-similar stochastic process, the increments of which on the time interval $\tau$ $\Delta X(\tau) = X(t + \tau) - X(t)$ have a Gaussian distribution:

$$P(\Delta X < x) = \frac{1}{\sqrt{2\pi}\sigma\tau^H} \int_{-\infty}^{x} \exp\left(\frac{-z^2}{2\sigma^2\tau^{2H}}\right) dz, \tag{2}$$

where $\sigma$ is the diffusion coefficient.

Exponent H determines the degree of long-range dependence of the fBm increments. The corresponding normalized correlation function is expressed as:

$$\text{Cor(t,s)} = 0.5 * (|t|^{2H} + |s|^{2H} - |t - s|^{2H}), \tag{3}$$

where t and s are time indices [28].

In general, for probabilistic forecasting, when Y is a real-valued response variable and X is a predictor variable, the goal is to find the conditional distribution function [31], which can be represented as

$$F(y \mid X = x) = P(Y \leq y \mid X = x). \tag{4}$$

Quantiles provide complete information about the distribution. For a continuous distribution function, the $\alpha$-quantile $Q\alpha(x)$ is defined as $Q\alpha(x) = \inf\{y: F(y \mid X = x) \geq \alpha\}$.

If the predictor variable is a time series

$$X(t) = \{X(1), X(2), \ldots, X(N)\}, \tag{5}$$

the task of probabilistic forecasting involves finding an algorithm that can predict the probability distribution of future values of the time series $X(t + 1), X(t + 2), \ldots, X(t + K)$, based on available past values X(t).

Thus, in the case of a fractional Brownian motion fBm time series X(t) of length N, with a Hurst exponent equal to H, the task is to investigate the properties of continuations

$$Xc(t) = \{X(N + 1), X(N + 2), \ldots, X(N + K)\}, \tag{6}$$

obtained using machine learning methods, such that

$$F(X(N + 1), X(N + 2), \ldots, X(N + K) \mid X(N), X(N - 1), \ldots, X(N - m)), \tag{7}$$

where F is the conditional distribution function, K is the forecast horizon, and m is the horizon of known values of the time series.

In each generated continuation Xc(t), the Hurst exponent H of the original time series X(t) should be preserved, as well as the scale (standard deviations of increments S). The quantiles of the continuations Xc(t) will be compared to the ground truth quantiles of possible continuations, and a metric assessing their discrepancies will be determined.

*2.2. Method of Creating the Evaluating Dataset*

There are several exact methods to simulate fBm realizations [25]. The Hosking method is simple, popular, and implemented in Python. The Hosking method is intended for the simulation of a stationary Gaussian sequence, which is increments of discrete-time fBm. An important feature of our research is that this method generates $X_{n+1}$ given $X_n, \ldots, X_1$ recursively.

We propose a method for creating a dataset of fBm time series and their ground truth continuations using the Hosking algorithm. Our method involves the parallel generation of M continuations Xc(t) for one original time series X(t) and the formation of a matrix of quantiles $Q\alpha = \{Q(\alpha i, tj)\}$ for each time step t. This dataset can then be used to evaluate probabilistic forecasting methods the for fBm time series.

Our method of creating a single example for the dataset for a specific Hurst exponent consists of the following steps:

1.  We use the Hosking method to generate values of the first N time steps of the predictor variable, which is a time series X(t) given by expression (5);
2.  Starting from the state of the internal variables of the Hosking algorithm after generating step N, we continue generating values for the next K time steps, obtaining a continuation Xc(t), defined by (6);
3.  We repeat step 2 M times, obtaining M different continuations from the original common starting point:

$$Xc_m(t) = \{Xc_m(1), Xc_m(2), \dots, Xc_m(K)\}, m = 1, 2, \dots, M; \tag{8}$$

4.  For each time step k, k = 1, 2, ... , K, we determine the quantiles of the values of M continuations, obtaining the desired quantile matrix;
5.  The dataset stores the target Hurst exponent, for example, X(t), the predictor variable X(t) of length N, and matrix $Q\alpha = \{Q(\alpha i, tj)\}$ of size 101× K, which contains, for each time step tj = 1, 2, ... , K, quantiles $\alpha i$, i = 1, 2, ... , 101 of the distribution of M continuations $\{Xc_m(t)\}$. The 0th and 1.0 quantiles represent the minimum and maximum values among the M variants at each time step tj.

This algorithm for creating a single example for the database can be used to populate the database with many examples for each of the various Hurst exponent values.

The example created using this method is illustrated in Figure 2. The figure displays a common row for the first 1024 time steps, followed by four colored curves representing the quantiles of the numerous possible continuations of the common beginning.

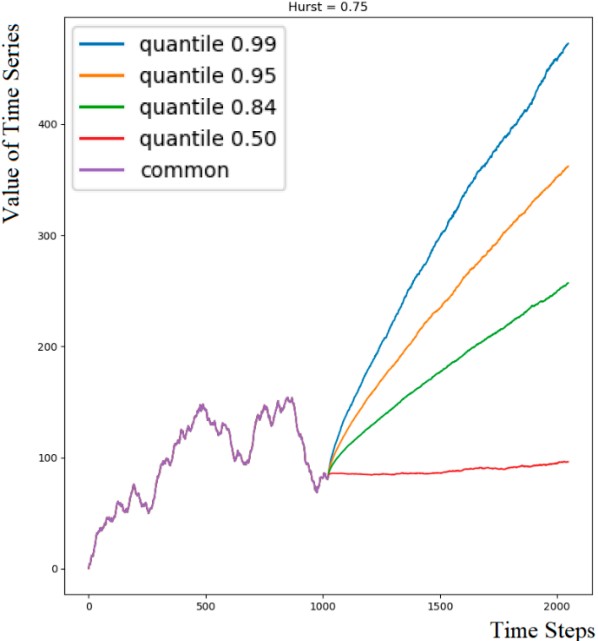

**Figure 2.** Visualization of an example of the proposed dataset. Common beginning of length 1024 and quantiles of its continuation length 1024 (quantiles 0.99, 0.95, 0.84, 0.50).

The code for generating a file of the dataset for a specific Hurst exponent can be found at the following link: https://www.kaggle.com/code/unfriendlyai/fbm-dataset-generator (accessed on 8 May 2023). Our fBm dataset is available on the Kaggle platform at: https://www.kaggle.com/datasets/unfriendlyai/fractal-brownian-motion (accessed on 8 May 2023).

*2.3. Predicting Quantiles of Time Series for Multiple Future Time Steps*

To iteratively predict quantiles for multiple future time steps using a one-step-ahead machine learning method, we propose the following approach:

1.  Train the machine learning model to output the distribution of the next time series value given the previous ones. The result can be in the form of parameters of a certain distribution, quantiles, or any other form that we can sample values from.
2.  Given a test example with a time series length of N X(t), the model predicts the distribution F represented by (7) of the next N + 1 value X(N + 1). Sample one value from the distribution F to obtain one possible next value of the continuation, Xc(1).
3.  Concatenate X(t) with Xc(t) and iteratively predict the next time step distribution, sampling one value for Xc(t + 1) for the next K time steps, obtaining a continuation Xc(t), defined by Equation (6).
4.  Repeat the second and third steps many times, obtaining many different continuations from the original single test example.
5.  Determine the quantiles of the values of the continuations, obtaining the desired quantiles for each time step.

This algorithm for estimating quantiles for future time steps of a single test example X(t) can be used to predict quantiles for many such examples (Figure 1). Then predicted quantiles can be compared with ground truth quantiles from the dataset to evaluate the efficiency of the probabilistic machine learning method. The same algorithm can be used to predict the probability of the value of an fBm time series exceeding a certain threshold after a specific number of time steps.

The advantage of using iterative one-step-ahead prediction over directly predicting multiple time steps is that one-step prediction is accessible to a wide array of machine learning methods, unlike predicting a sequence of future values. Furthermore, once the model is trained, it can be used to predict an arbitrary number of future time steps.

*2.4. Assessment of Accuracy and Reliability of Continuations*

In our research, it is essential to compare the properties of the obtained continuations Xc(t) with the properties of the original time series X(t).

There are several properties we need to consider for comparison: the Hurst exponent H, the normality of the distribution, and the magnitude of increments. If the values of H, the standard deviation of the increments, and the normality of the distribution coincide, we can conclude that the obtained continuations are realizations of the fractional Brownian motion with the same H as the original ones.

The aforementioned properties are examined and verified through the following procedures:

1.  Calculating the Hurst exponent (H) using the Whittle estimator demonstrated the best accuracy for time series with lengths of 512 and above, which we plan to evaluate, as shown in a comparative study [32]. This allows us to assess whether the fractal properties of the original time series X(t) are preserved in the generated continuations Xc(t).
2.  Determining if the increments are normally distributed using the Anderson–Darling test. This test is a statistical method used to check whether a given sample of data follows a specific distribution (in our case, normal distribution). If the increments are normally distributed, it indicates that the generated continuations Xc(t) maintain the same distribution law as the original time series X(t).
3.  Comparing the standard deviations of increments (S) between the original time series X(t) and the generated continuations Xc(t). By ensuring that the standard deviations of increments are preserved, we can confirm that the scale of the generated continuations Xc(t) is consistent with the original time series X(t).

In addition to the requirement that the generated continuations Xc(t) must have the same fractal and statistical properties as the original series X(t), we need to assess the

quality of the forecasting. Ideally, for each example, X(t) with its corresponding value of H, the matrix of quantiles of the continuations $Qc(\alpha_i)$ should match the true matrix $Q(\alpha_i)$ presented in the dataset. As a measure of deviation Qdev, we will use the absolute difference between the true and predicted quantile values at the given time step tj, normalized by the difference between the true value of the quantile $Q(\alpha_i)$ and the adjacent $Q(\alpha_{i+1})$ for each quantile from 0.01 to 0.99 and each time step, averaging it (Figure 3):

$$Qdev = \frac{1}{99K} \sum_{j=1}^{K} \sum_{i=1}^{99} \frac{|Qc(\alpha i, tj) - Q(\alpha i, tj)|}{Q(\alpha i+1, tj) - Q(\alpha i, tj)}, \tag{9}$$

where $QC(\alpha i, tj)$ is predicted $\alpha i$-quantile values at the time step tj; $Q(\alpha i, tj)$ is true $\alpha i$-quantile values at the time step tj; K is the length of the continuation.

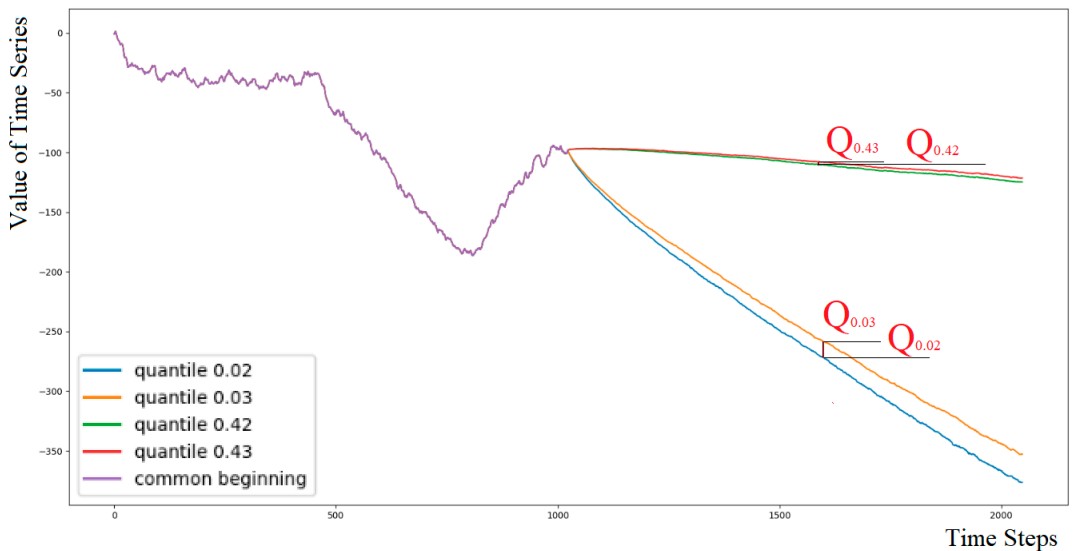

**Figure 3.** The figure illustrates the varying distances between adjacent quantiles for different time steps and distances from the median. To interpret the deviation Qdev in the "inter-quantile" measure, normalization is performed by dividing absolute deviation by the difference between adjacent quantiles.

This interpretable metric provides an intuitive understanding of the model's performance by expressing the error in the context of the distance between adjacent quantiles. A lower value for this metric indicates that the predicted quantiles are closer to the true quantiles, making the model's forecasts more accurate.

The metric Qdev characterizes the deviation of the time series continuation Xc(t) for a single example X(t) in the dataset for a specific value of H. Since the values of Qdev depend on random time realizations, it is necessary to average over all calculated examples with the same Hurst exponent to improve accuracy and reliability:

$$QDmean = \frac{1}{Nexample} \sum_{i=1}^{Nexample} Qdev(i) \tag{10}$$

where Qdev is the metric for a single example in the dataset, and Nexample is the number of examples in the dataset.

By performing the above-described checks, we can validate the quality of the generated continuations and ensure that our method effectively captures the essential fractal and statistical properties of the original time series X(t).

### 2.5. Machine Learning Methods for Probabilistic Forecasting

According to reviews [31,33,34], there are two broad classes of probabilistic methods: those predicting the parameters of a predefined distribution shape, and those addressing

the problem when the distribution shape is not specified. Although the first class of methods is perfectly suitable for predicting fractional Brownian motion, this research aims to explore methods that are not limited by a predefined probability distribution shape and are applicable to studying complex fractal time series.

For this study, we used implementations of the methods listed in Table 1.

**Table 1.** Probabilistic machine learning methods and corresponding implementations.

| Method | Output | Implementation |
|---|---|---|
| Natural gradient boosting NGBoost [17] | Estimates of the mean and standard deviation of the normal distribution | Python package ngboost |
| Catboost with Uncertainty (CBU) | Estimates of the mean and standard deviation of the normal distribution | Python package catboost |
| Mean, location, scale, and shape (LSS) XGBoostLSS and LightGBMLSS [18,19] | Estimates of the mean and standard deviation of the normal distribution | Alexander März's (StatMixedML) repository on github |
| Nonparametric Probabilistic Regression (PRESTO) [24] | Distribution without predefined shape | Our implementation of the author's idea using CatBoost, LightGBM, and Scikit-learn's Logistic Regression |
| Training a separate model for each quantile | Quantiles predictions | Python packages catboost, lightgbm |
| Quantile regression (QuantReg) [23] | Quantiles predictions | Python package statsmodels |
| Random Forest Quantile Regression [22] | Quantiles predictions | Python package sklearn_quantile |

By comparing these methods, we aim to find the most suitable technique for modeling and forecasting fractal time series data while preserving the essential fractal properties. This will help us to better understand the advantages and limitations of different probabilistic prediction methods in the context of complex, long-range dependent, and self-similar time series data, ultimately leading to more accurate and reliable forecasts for various practical applications.

The code and results of our experiments can be accessed at https://www.kaggle.com/datasets/unfriendlyai/fractal-brownian-motion/code (accessed on 8 May 2023).

## 3. Results

### 3.1. Dataset

3.1.1. Training Dataset

In a supervised learning setting, a machine learning model requires a predictor and a ground truth target. Probabilistic forecasting methods use the same training datasets as point-predicting methods. For time series, the training process for each algorithm involves learning to find the distribution of the (n + 1)-th element in the series based on the known n previous values, according to Formula (7).

To train models, there is no need to store the dataset, as it is possible to generate millions of fBm examples with specified Hurst exponent within a few minutes. We used the Hosking method for Hurst exponent values starting from 0.9 and the Davies–Harte method [25] for the Hurst exponent values below 0.9.

3.1.2. Evaluating Dataset

The dataset was created with the following parameters:

- Number of files—99, each for the value of H from 0.01 to 0.99 with a step of 0.01;
- Number of records in each file—50;
- The length of the original time series N—1024;

- The length of the continuations of the original series, for which quantiles are provided K—1024;
- The number of continuations of the time series to calculate quantiles M—10,001;
- The number of quantiles for each time step is 101 (the 0th and 1.0th quantiles are equal to the maximum and minimum values among M examples);
- Time series increments are normalized (divided by STD of increments); the standard deviations of increments for all examples are one;
- Original time series are presented as cumulative sums of increments. Quantiles are calculated on their cumulative continuations.

Each file, which consists of 50 examples of a certain Hurst exponent mentioned in the filename, takes nearly 2 h to create.

For experiments, we use dataset examples for a range of Hurst exponent values to represent anti-persistent, persistent, and near-independent fBm examples: 0.30, 0.35, 0.45, 0.53, 0.60, 0.65, 0.72, 0.85, 0.90, and 0.93.

### 3.2. Experiment Setting

The experiment was conducted in two stages:

1. Feeding the entire time series of length 1024 from the dataset to the machine learning model and extending it by 1024 steps, allowing for the evaluation of the Hurst exponent of the continuations and the model's ability to reliably extend the series for a large number of time steps. Since the dataset was prepared using Hosking's algorithm, and the series is included entirely starting from the zero value and the first increment obtained by the algorithm, the models receive complete data in this case. The series depends only on itself and does not depend on past increments unknown to the model.

2. Feeding only the last 512 values out of the 1024 available in the dataset to the machine learning model. In this case, the true values depend on some unknown past that the model is not aware of. We extend the series by only 64 steps, which significantly reduces the computation time and allows for obtaining performance estimates of the models when they are not constrained by strict time frames.

Since time series have fractal properties, providing all 1024 (or 512) values as input to the models may not be efficient in terms of memory and computational performance constraints. Given the importance of the nearest dependency, it is reasonable to input a certain number of the most recent values (e.g., 8–16) in their entirety, while selecting every 8th (or every 16th, 32nd, or 64th, depending on the model's memory requirements) value from the remaining length of the series.

### 3.3. Investigating the Properties of Continuations
### 3.3.1. Calculating the Hurst Exponent of Continuations

All continuations of the time series were evaluated using the Whittle method to determine the Hurst exponent and averaged. The results are presented in Table 2. For continuations of length 1024, the estimated Hurst exponent values were almost identical to the true values of the Hurst exponent for the original time series. The only exception is LightGBMLSS with the Gaussian_AutoGrad distribution when extending the series with the highest Hurst exponent H = 0.93 (in this case, all other indicators were also unsatisfactory).

As can be seen from the table, the lowest accuracy between the Hurst exponent of the original series and the predicted continuation was observed at high H values (0.90 and 0.93). At high Hurst exponent values (H > 0.85), the Hurst exponent values of the continuations deviate more significantly from the original time series values. This suggests that the methods may have more difficulty in capturing the long-range dependence of time series with higher Hurst exponent values.

**Table 2.** Comparison of Hurst Exponent Accuracy at different H Values of original time series. * The red mark signifies an outlier, a data point that deviates significantly from other observations, and the marking was done intentionally for emphasis.

| Method | H = 0.3 | H = 0.35 | H = 0.45 | H = 0.53 | H = 0.6 | H = 0.65 | H = 0.72 | H = 0.85 | H = 0.9 | H = 0.93 |
|---|---|---|---|---|---|---|---|---|---|---|
| CBU 500 iter. | 0.299 | 0.349 | 0.451 | 0.529 | 0.6 | 0.649 | 0.718 | 0.848 | 0.895 | 0.935 |
| CBU 1000 iter. | 0.299 | 0.351 | 0.449 | 0.529 | 0.599 | 0.649 | 0.719 | 0.848 | 0.897 | 0.935 |
| LightGBMLSS Gaussian | 0.298 | 0.349 | 0.453 | 0.528 | 0.598 | 0.648 | 0.711 | 0.843 | 0.895 | 0.926 |
| LightGBMLSS Gaussian_AutoGrad | 0.297 | 0.349 | 0.451 | 0.531 | 0.602 | 0.654 | 0.713 | 0.849 | 0.895 | 0.83 * |
| XGBoostLSS Gaussian | 0.301 | 0.356 | 0.447 | 0.534 | 0.597 | 0.651 | 0.719 | 0.844 | 0.9 | 0.922 |
| PRESTO with CatBoost | 0.298 | 0.349 | 0.449 | 0.529 | 0.598 | 0.648 | 0.718 | 0.846 | 0.894 | 0.925 |
| PRESTO with LightGBM | 0.302 | 0.352 | 0.452 | 0.527 | 0.595 | 0.644 | 0.713 | 0.841 | 0.889 | 0.922 |
| PRESTO with Logistic Regression | 0.299 | 0.35 | 0.448 | 0.529 | 0.599 | 0.649 | 0.719 | 0.848 | 0.897 | 0.928 |
| CatBoost for each quantile | 0.306 | 0.357 | 0.454 | 0.525 | 0.593 | 0.643 | 0.713 | 0.84 | 0.886 | 0.914 |
| LightGBM for each quantile | 0.306 | 0.357 | 0.454 | 0.525 | 0.593 | 0.644 | 0.713 | 0.839 | 0.886 | 0.916 |

### 3.3.2. Determining If the Increments Are Normally Distributed

The continuations obtained by all methods successfully passed the Anderson–Darling test [26] across the entire range of Hurst exponent values, except for CBU and LightGBMLSS with the Gaussian_AutoGrad distribution on high H values greater than 0.85.

### 3.3.3. Checking the Standard Deviation of the Increments

Another important aspect of evaluating the performance of the considered methods is to determine if the increments in the generated time series continuations Xc have a standard deviation equal to that of the original time series X, which in our case is equal to 1. This is crucial because it helps ensure that the methods maintain the same level of variability in the generated time series as in the original data.

All methods successfully extend the original time series, in which the standard deviation (SD) of the increments is equal to one, with continuations having SD of increments very close to one for Hurst exponent values below 0.85. For Hurst exponent values starting from 0.85, all methods show a decrease in the SD of increments, which is most noticeable at H = 0.93. In this case, CBU and LightGBMLSS with the Gaussian_AutoGrad distribution unsatisfactorily lower the SD to 0.2–0.4, while the other methods have SD of increments above 0.75 (Table 3).

This demonstrates that the two aforementioned methods (CBU and LightGBMLSS with the Gaussian_AutoGrad distribution) are the least adapted for extending fractal time series with long-range dependencies, while the other methods have relative issues only at the extreme Hurst exponent value of 0.93.

### 3.4. Investigating Forecasting Accuracy When Extending a Time Series of Length 1024 by 1024 Time Steps

In this experiment, when using the full evaluation dataset, not all methods were able to show results, as NGBoost, PGBM (even when using GPU), IBUG, Quantile Regression QuantReg (statsmodels package), and the quantile-forest package for Quantile Regression Forests turned out to be too slow to train sufficiently for comparable results.

We determine the deviation of forecasted quantiles Qdev using the "inter-quantile" measure (9). Lower values indicate better performance. Some methods provide high results only for H = 0.53 (i.e., very close to 0.5). These methods are LightGBMLSS Gaussian_AutoGrad and CBU (Table 4). However, when long-range dependencies are present in

the time series, their results are significantly worse compared to other methods. Therefore, it is concluded that such methods are not suitable for the analysis of fractal time series.

**Table 3.** The standard deviation of increments of continuations (expected to be close to 1). * The red mark signifies an outlier, a data point that deviates significantly from other observations, and the marking was done intentionally for emphasis.

| Method | H = 0.3 | H = 0.35 | H = 0.45 | H = 0.53 | H = 0.6 | H = 0.65 | H = 0.72 | H = 0.85 | H = 0.9 | H = 0.93 |
|---|---|---|---|---|---|---|---|---|---|---|
| CBU 500 iter. | 0.956 | 0.974 | 0.996 | 0.998 | 0.986 | 0.965 | 0.921 * | 0.75 * | 0.573 * | 0.418 * |
| CBU 1000 iter. | 0.955 | 0.973 | 0.995 | 0.997 | 0.985 | 0.965 | 0.919 * | 0.753 * | 0.578 * | 0.414 * |
| LightGBMLSS Gaussian | 0.996 | 0.997 | 1.003 | 0.998 | 1.002 | 0.996 | 0.997 | 0.975 | 0.892 | 0.778 |
| LightGBMLSS Gaussian_AutoGrad | 0.995 | 0.996 | 0.996 | 1.001 | 1.002 | 0.996 | 0.998 | 0.981 | 0.902 | 0.274 * |
| XGBoostLSS Gaussian | 0.996 | 0.998 | 0.998 | 0.998 | 0.998 | 0.997 | 0.995 | 0.976 | 0.899 | 0.776 |
| PRESTO with CatBoost | 1 | 1.001 | 1.001 | 1.001 | 1.002 | 1 | 0.999 | 0.986 | 0.897 | 0.786 |
| PRESTO with LightGBM | 1.001 | 1.002 | 1.002 | 1.002 | 1.002 | 1.001 | 0.999 | 0.978 | 0.903 | 0.795 |
| PRESTO with Logistic Regression | 1.001 | 1.001 | 1.002 | 1.002 | 1.002 | 1.001 | 1 | 0.977 | 0.906 | 0.789 |
| CatBoost for each quantile | 1.004 | 1.004 | 1.004 | 1.004 | 1.005 | 1.003 | 1.002 | 0.983 | 0.905 | 0.809 |
| LightGBM for each quantile | 1.004 | 1.004 | 1.004 | 1.004 | 1.004 | 1.003 | 1.002 | 0.983 | 0.907 | 0.814 |

**Table 4.** The deviation of forecasted quantiles Qdev when extending a time series of length 1024 by 1024 time steps. * The red mark signifies an outlier, a data point that deviates significantly from other observations, and the marking was done intentionally for emphasis. The boldface used to highlight the models with the best performance in the table.

| Method | H = 0.3 | H = 0.35 | H = 0.45 | H = 0.53 | H = 0.6 | H = 0.65 | H = 0.72 | H = 0.85 | H = 0.9 | H = 0.93 |
|---|---|---|---|---|---|---|---|---|---|---|
| CBU 500 iter. | 1.854 | 1.331 | 0.741 | **0.544** | 1.339 | 1.614 | 2.805 | 6.659 * | 10.597 * | 14.594 * |
| CBU 1000 iter. | 1.803 | 1.271 | 0.71 | 0.963 | 1.127 | 1.604 | 3.02 | 6.289 * | 10.706 * | 14.917 * |
| LightGBMLSS Gaussian | 1.405 | 1.175 | 0.914 | 1.256 | **1.077** | 2.134 | 3.142 | 2.74 | 4.477 | 6.573 |
| LightGBMLSS Gaussian_AutoGrad | 1.44 | 1.24 | 1.113 | 0.594 | 1.223 | 2.514 | 1.971 | 3.741 | 4.594 | 36.382 |
| XGBoostLSS Gaussian | 1.493 | 1.174 | 0.955 | 0.861 | 1.236 | 1.676 | 1.975 | 11.028 * | 4.49 | 6.806 |
| PRESTO with CatBoost | 1.606 | 1.557 | 1.307 | 1.668 | 2.035 | 1.669 | 2.355 | 3.976 | 4.609 | 6.925 |
| PRESTO with LightGBM | 1.62 | 1.408 | 0.975 | 1.058 | 1.512 | 1.579 | 1.99 | 3.703 | 5.651 | 8.709 |
| PRESTO with Logistic Regression | **1.319** | **1.024** | **0.647** | 0.729 | 1.85 | **1.174** | **1.41** | **1.856** | **3.361** | **5.863** |
| CatBoost for each quantile | 2.338 | 1.978 | 1.26 | 1.334 | 1.602 | 2.747 | 2.964 | 7.037 * | 5.984 | 9.793 |
| LightGBM for each quantile | 2.101 | 1.846 | 1.059 | 0.923 | 1.536 | 3.014 | 3.767 | 4.742 | 5.961 | 9.713 |

An important result of the study is that Nonparametric Probabilistic Regression with Coarse Learners (PRESTO) [24] demonstrated comparable performance with the most successful LSS method (XGBoostLSS and LightGBMLSS with Gaussian distribution). The author of PRESTO primarily considered the application of decision trees as a classifier in his method. As the core idea does not limit us in the choice of classifier, we proposed the use of logistic regression from the sklearn package. Interestingly, this simplest approach turned out to be the most accurate in this experiment.

All methods were generally more accurate for Hurst exponent values below 0.85 (Figure 4).

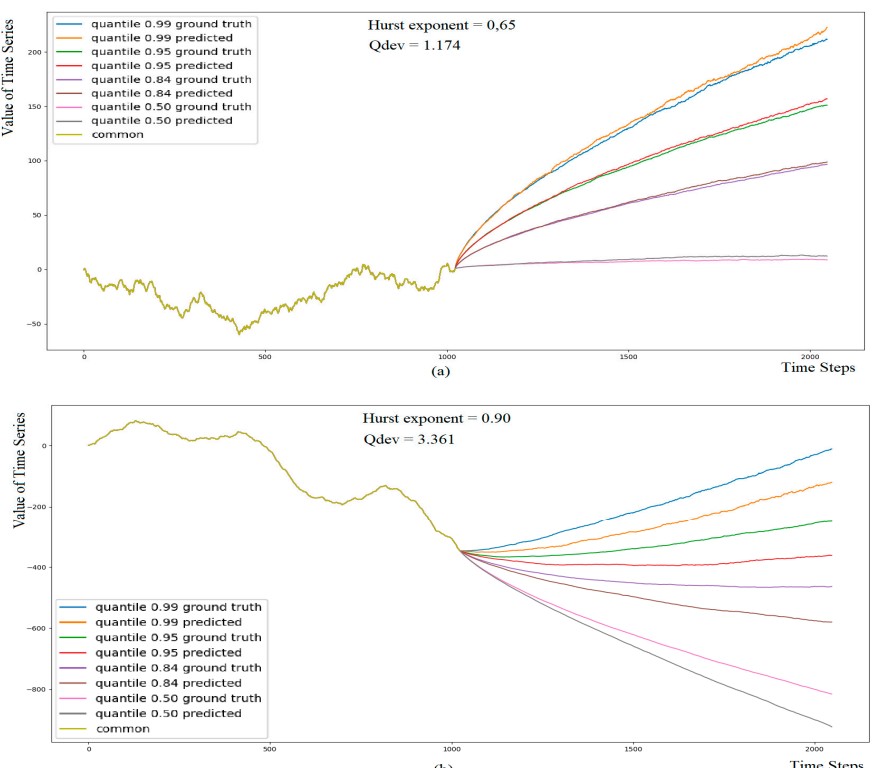

**Figure 4.** Quantile prediction was produced using PRESTO with the Logistic Regression method for Hurst exponent values 0.65 (**a**) and 0.90 (**b**). At high Hurst exponent values (**b**) the predicted quantiles of the continuations deviate more significantly from the ground truth quantiles.

### 3.5. Investigating Forecasting Accuracy When Extending a Time Series of Length 512 by 64 Time Steps

In this study, when providing only the last 512 values out of the 1024 available in the dataset, NGBoost showed very high results only for H = 0.53 (i.e., very close to 0.5), significantly lagging behind the leaders in other Hurst exponent values. A possible reason is the lack of features due to time constraints for this relatively slow method. Thus, this method may have potential, but its slowness is a significant obstacle to its use compared to other methods.

As in the previous study, CBU showed unsatisfactory results in capturing the long-range dependence of time series with higher Hurst exponent values. In addition, in the entire range except for H = 0.53, it significantly lagged behind the LSS method, modeled Gaussian distribution only, and is not attractive for the study of fractal time series.

Among the parametric methods, our choice is for the LSS method (XGBoostLSS is better when H > 0.6, and LightGBMLSS with Gaussian distribution is better when H < 0.6). The most significant interest is in the results of nonparametric methods.

The results are listed in Table 5.

### 3.6. Comparing Forecasting Accuracy Given Time Series of Length 1024 and 512

In this study, we compared the performance of the most successful model, PRESTO with Logistic Regression, when extending time series of different lengths. In all cases, the last 16 values of the time series were used as input, while the rest of the input varied between the experiments. The first experiment used one out of every 32 values of the original time series with a length of 1024, while the second experiment used one out of every 32 values of the first 512 values of the time series. The third experiment fed one out of every 16 values of the first 512 values to align the number of features. The results are listed in Table 6.

**Table 5.** The deviation of forecasted quantiles Qdev when extending a time series of length 512 by 64 time steps. The boldface used to highlight the models with the best performance in the table.

| Method | H = 0.3 | H = 0.35 | H = 0.45 | H = 0.53 | H = 0.6 | H = 0.65 | H = 0.72 | H = 0.85 | H = 0.9 | H = 0.93 |
|---|---|---|---|---|---|---|---|---|---|---|
| NGBoost | 2.576 | 1.756 | 1.02 | 0.8 | 1.386 | 1.782 | 2.168 | 4.163 | 4.19 | 7.863 |
| CBU | 2.178 | 1.966 | 1.984 | 1.642 | 2.146 | 2.177 | 2.036 | 6.938 | 10.688 | 13.962 |
| XGBoostLSS Gaussian | 1.123 | 0.838 | 0.781 | 0.831 | 1.207 | 1.176 | **1.025** | 2.115 | 2.751 | 6.001 |
| LightGBMLSS Gaussian | 1.111 | 1.066 | 0.589 | 0.599 | 1.388 | 1.042 | 1.068 | 4.607 | 3.203 | 32.828 |
| PRESTO with CatBoost | 1.499 | 1.174 | 1.565 | 1.49 | 1.794 | 1.219 | 1.576 | 1.977 | 3.556 | 6.048 |
| PRESTO with LightGBM | 0.995 | 0.804 | 0.727 | 0.767 | 0.774 | 1.034 | 1.077 | 1.895 | 3.022 | 5.783 |
| PRESTO with Logistic Regression | **0.94** | 0.747 | **0.505** | **0.502** | **0.669** | 0.905 | 1.077 | 1.626 | **2.747** | 5.839 |
| LightGBM for each quantile | 1.165 | 1.022 | 0.618 | 0.798 | 1.045 | 1.013 | 1.262 | 1.931 | 2.971 | 5.592 |
| CatBoost for each quantile | 1.318 | 1.178 | 0.764 | 0.795 | 1.02 | 1.296 | 1.831 | 2.789 | 4.256 | 8.73 |
| QR statsmodels | 1.096 | **0.74** | 0.689 | 0.611 | 0.769 | **0.878** | 1.135 | **1.467** | 2.961 | **5.53** |
| Random Forest QR | 1.629 | 1.502 | 1.55 | 1.639 | 1.307 | 1.579 | 1.666 | 5.672 | 4.13 | 6.45 |

**Table 6.** Comparison of forecasting quantiles using different lengths of the predictor.

| Features | H = 0.3 | H = 0.35 | H = 0.45 | H = 0.53 | H = 0.6 | H = 0.65 | H = 0.72 | H = 0.85 | H = 0.9 | H = 0.93 |
|---|---|---|---|---|---|---|---|---|---|---|
| 1/32 of 1024 | 0.659 | 0.648 | 0.505 | 0.656 | 0.66 | 0.56 | 0.544 | 0.86 | 2.394 | 5.204 |
| 1/16 of 512 | 0.986 | 0.702 | 0.527 | 0.537 | 0.678 | 0.858 | 1.031 | 1.601 | 2.915 | 5.521 |
| 1/32 of 512 | 1.036 | 0.722 | 0.551 | 0.466 | 0.633 | 0.949 | 1.021 | 1.79 | 3.019 | 5.348 |

The results of this study show that the length of the input time series plays an important role when there are long-term dependencies (Hurst exponent significantly different from 0.5). When only half of the time series was fed to the model, it significantly underperformed in generating continuations, even when values were sampled more densely (every 16th value compared to every 32nd value). However, when the Hurst exponent was close to 0.5 (indicating an almost independent time series), the situation was reversed. Extra features from a longer time series were irrelevant and only worsened the prediction.

## 4. Discussion

By employing iterated approaches with one-period-ahead models, we can effectively generate numerous time series continuations that preserve the fractal properties of the original time series. This enables us to evaluate the performance of our model in capturing the complex, long-range dependencies, and self-similarity exhibited by fractal time series data. At the same time, it is essential to assess the quality of the continuations—whether they have the same measurable fractal or statistical characteristics—to guarantee the model's reliability. In turn, this allows for a more comprehensive assessment of the model's ability to provide accurate and reliable forecasts for time series with fractal properties, which is essential for various practical applications in fields such as finance, meteorology, and environmental studies.

The results suggest that machine learning methods can learn the distribution of the next increment of the time series, taking into account not only long-range dependence but also preserving the numerical values of the measurable indicator of this long-range dependence, namely, the Hurst exponent (Table 2). This provides a perspective that machine learning methods can be used not only for predicting quantiles of the distribution but also for generating plausible time series while preserving fractal and statistical properties.

Leveraging the fractal properties of time series data in this way allows for a reduction in computational demands and an increase in the training set size by reducing the number of features. The last 16 values of the time series were fed to the investigated model in their

entirety, and to account for long-term dependencies, only every 8th (16th, 32nd) value of the remaining series was sufficient to obtain accurate predictions (Table 6).

Fractional Brownian motion time series exhibit standard (Gaussian) distribution of increments, making them ideally suited for forecasting using methods that assume a conditional distribution for the target variable, such as a normal distribution where both mean (μ) and standard deviation (σ) depend on the explanatory variables (X). It was anticipated that these methods would outperform nonparametric methods in forecasting tasks. The performance of parametric methods was investigated in comparison to nonparametric methods, providing valuable insights into the potential limitations of nonparametric approaches. Furthermore, it will enable practitioners to assess the risks associated with utilizing nonparametric methods for analyzing time series data exhibiting fractal properties when forecasting the distribution shape is challenging or not feasible.

It was found that not all parametric probabilistic methods perform well across the entire range of Hurst exponent values. The nonparametric PRESTO method proved to be comparable in effectiveness to parametric methods, while also being applicable for predicting time series with fractal characteristics whose probability distribution shape cannot be predetermined.

The main inspiring question of the study was to what extent parametric methods outperform nonparametric ones. Unexpectedly, the latest nonparametric method PRESTO showed itself quite acceptable, having the potential to study time series with any complex probability distribution of the next increment, and especially the ability to conduct detailed research and modeling of heavy-tailed distributions. Not to mention that it surpassed most parametric methods in terms of speed and accuracy.

This result highlights the potential of nonparametric methods such as PRESTO in the analysis and modeling of fractal time series, especially when the underlying probability distribution is unknown or complex. The flexibility and adaptability of nonparametric methods make them suitable for a wider range of applications and could lead to more accurate predictions and insights in various fields where fractal time series are relevant.

While parametric methods such as LSS (XGBoostLSS and LightGBMLSS with Gaussian distribution) showed promising results, the nonparametric PRESTO demonstrated that it could compete with these methods and even outperform them in certain situations. This finding could encourage further development and research into nonparametric methods, leading to more advanced and versatile tools for analyzing and modeling fractal time series.

The best performance was demonstrated by the PRESTO method with Logistic Regression as a classifier. The Quantile Regression method, QuantReg from the statsmodels package, also showed strong performance. This may be due to the fact that the time series was synthesized, meaning that it did not contain any outliers. Increments are normally distributed, and the task turned out to be perfectly suited for linear methods. In the case of real-world time series, other methods capable of capturing non-linear dependencies may demonstrate better performance.

In our analysis, we observed a specific trend. When the Hurst exponent is close to 0.5, the majority of models predict well. However, when the exponent is 0.85 or higher, the performance of all models deteriorates, some simply fail. This tendency is an indicative marker of the models' quality, separating those that are effective from those that are not. The higher the Hurst exponent, the more challenging the prediction becomes. This is primarily because certain models struggle to capture long-term dependencies. While this is a challenge for all models, some particularly struggle. Thus, further research is needed to develop models that are capable of capturing long-term dependencies, especially in time series with high Hurst exponent values.

Parametric LSS methods (XGBoostLSS and LightGBMLSS with Gaussian distribution), demonstrated good performance in capturing long-range dependence in time series and extending them while maintaining Hurst exponent values. These methods were generally more accurate for Hurst exponent values below 0.85 (H < 0.85).

The nonparametric method PRESTO showed promising results, competing with the best parametric methods in terms of accuracy and speed. Its flexibility in modeling complex and unknown probability distributions makes it a suitable choice for a wide range of applications and indicates potential for further development.

Some methods, such as CBU, LightGBMLSS with Gaussian_AutoGrad, NGBoost, and PGBM, showed limitations in capturing long-range dependence in fractal time series, especially for high Hurst exponent values (>0.85). These methods may not be suitable for analyzing fractal time series or may require further refinement.

The best performance was demonstrated by the simplest methods, namely, PRESTO with Logistic Regression and Quantile Regression QuantReg from the statsmodels package.

## 5. Conclusions

This study aimed to evaluate the performance of various parametric and nonparametric probabilistic forecasting machine learning methods in extending the fBm time series with different Hurst exponent values. Self-similar properties of fBm time series have been reliably reproduced in the continuations of the time series predicted by machine learning methods. The values of the Hurst exponent have been preserved in the generated continuations. The increments of the continuations have had a normal distribution and their standard deviation has coincided with that of the original ones.

Overall, this study demonstrates that probabilistic forecasting machine learning methods allow the prediction of the dynamics of fractal time series while preserving their self-similarity and statistical characteristics. Furthermore, we propose a distinctive and effective approach to assess the performance of various probabilistic forecasting methods in predicting the fBm time series. This methodology involving the use of the Hosking algorithm for generating ground truth quantiles of possible continuations is novel.

The best performance was demonstrated by the method that converts the probabilistic regression problem into one of multi-classification—PRESTO, with Logistic Regression proving to be the most successful classifier. Additionally, the Quantile Regression method, QuantReg from the statsmodels package, also showed strong performance.

Further research and development in nonparametric methods could lead to more advanced and versatile tools for modeling and predicting fractal time series in various fields. The proposed approach enables the consideration of long-term dependencies and scaling properties of the data, leading to more accurate forecasts and a deeper understanding of the dynamics and structure of the investigated processes. The presented machine learning methods can be applied to various domains, including financial market forecasting, disease dynamics, seismic activity, telecommunication traffic load, and others.

**Author Contributions:** Conceptualization, methodology, investigation, validation, writing—original draft preparation, writing—review and editing, L.K. and R.L.; software, resources, data curation, visualization, R.L.; formal analysis, supervision, project administration, L.K. All authors have read and agreed to the published version of the manuscript.

**Funding:** This research received no external funding.

**Data Availability Statement:** The data presented in this study are openly available here: https://www.kaggle.com/datasets/unfriendlyai/fractal-brownian-motion (accessed on 8 May 2023).

**Conflicts of Interest:** The authors declare no conflict of interest.

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
