# Peer review of "Probabilistic Machine Learning Methods for Fractional Brownian Motion Time Series Forecasting"

_fractalfract, doi:10.3390/fractalfract7070517_

Round 1
Reviewer 1 Report
In this paper under review, the authors propose "to evaluate the performance of various parametric and non-parametric probabilistic forecasting machine learning methods in extending fractional Brownian motion (fBm) time series with different Hurst exponent values."
The probabilistic forecasting of trajectories for a self-similar process has been used and a specific approach to define FBm was adopted. The apllied methodology and the properties of the study seems to be analyzed rigorously.
Experiments have been conducted to test and compare the performance and authors state that the best performance was demonstrated by the simplest methods, namely PRESTO with Logistic Regression and Quantile Regression QuantReg from the stats models package.
The paper is interesting. I have just some comments/suggestions to the authors.
1. 'Abstract' and 'Introduction' section are clear and emphasize the relevance of the achieved results for the area and the main findings and contribution of this work.
2. Section 2 (Materials and Methods) intend to present the state-of-the-art through citations of published works. Nonetheless, I think that an error must have ocurred while papers that have been cited along the text did not appear in a list of References at the end of the manuscript (in the pdf version). Please check about this and include references cited in the text in the list of References.
3. Why this specific approach (adopted in this work) was chosen (to the detriment of others) to define FBm? Please clarify to the reader.
4. It seems that the machine learning methods explored in this work fail in capturing the long-range dependence for time series with higher Hurst exponent values. Why does this happen? I suggest including more details regarding these limitations.
5. Although the authors have proposed different probabilistic machine learning methods in this study and presented a performance comparison against them, a deeper discussion related to the achieved results still lack and it should be provided.
6. I suggest to improve the statistical evaluation of the time series continuation for a specific value of H. For example, one more metric could be added to the metric Qdev adpted. This should strength this work.
7. The manuscript needs proofreading for minor typos fixes.
In my view, the paper is interesting and can contribute to the literature after major revision.
In my view, a moderate editing of English language is required.
Author Response
Dear Reviewer,
We would like to express our sincere gratitude for taking the time to review and valuable comments, which has greatly contributed to improving the quality of our article.
Please see the attachment.

Reviewer 2 Report
Machine learning methods for probabilistic forecasting of fractional Brownian motion (fBm). NGBoost, PRESTO, QuantReg and other methods are implemented in python and links for used codes are given. A metric that characterizes the deviation of the time series continuation is defined and Standard deviation of increments of continuations is computed. Accuracy of forecasting is studied for different lengths and steps of the time series. Conclusions upon the Hurst index influence are made.
Please complete the references cited.
Author Response
Dear Reviewer,
We would like to express our sincere gratitude for taking the time to review our manuscript. We want to inform you that we have made significant additions to the list of cited references in response to your comment ([11-13, 27-31]).
Reviewer 3 Report
Here are some comments to consider:
-There are paragraphs in which the writing can be improved.
-It remains to explain the meaning of the parameters of the equations.
-Figures can be upgraded. Include axis information.
-Check the values in red in tables 3 and 4.
-Although the approach is interesting, it is recommended to validate it with experimental results of a physical phenomenon.
None
Author Response

(The authors gave the same response as above.)

Reviewer 4 Report
This paper can be accepted.

Author Response
Dear Reviewer,
We would like to express our sincere gratitude for taking the time to review and valuable comments, which has greatly contributed to improving the quality of our article.
Please see the attachment

Round 2
Reviewer 1 Report
I would like to point out that authors have addressed my comments/questions.
In my view, the manuscript can be accepted now.
In my opinion, the moderate English language is required.
Reviewer 3 Report
The authors have corrected the manuscript.